# THE *Entropy* MECHANISM OF REINFORCEMENT LEARNING FOR REASONING LANGUAGE MODELS

## ABSTRACT

This paper aims to overcome a major obstacle in scaling reinforcement learning (RL) for reasoning with large language models (LLMs), namely the collapse of policy *entropy*. Such phenomenon is consistently observed across vast RL runs without entropy intervention, where the policy entropy dropped sharply at the early training stage, leading to an overly confident policy model. As a consequence, this diminished exploratory ability is always accompanied with the saturation of policy performance. In practice, we establish a transformation equation $R = -a \exp \mathcal{H} + b$, between entropy $\mathcal{H}$ and downstream performance $R$, where $a, b$ are fitting coefficients. This empirical law strongly indicates that, the policy performance is traded from policy entropy, thus bottlenecked by its exhaustion, and the ceiling is fully predictable ($\mathcal{H} = 0, R = -a + b$). Our finding necessitates entropy management for continuous exploration toward scaling compute for RL. To this end, we investigate entropy dynamics both theoretically and empirically. Our derivation highlights that, the change in policy entropy is driven by the covariance between action probability and the change in logits, which is proportional to its advantage when using Policy Gradient-like algorithms (Williams, 1992). For example, a high-probability action with high advantage would reduce policy entropy, while a rare action with high advantage would increase policy entropy. Empirical study shows that, the values of covariance term and entropy differences matched exactly, supporting the theoretical conclusion. Moreover, the covariance term stays mostly positive throughout training, further explaining why policy entropy would decrease monotonically. Through understanding the mechanism behind entropy dynamics, we motivate to control entropy by restricting the update of high-covariance tokens. Specifically, we propose two simple yet effective techniques, namely `Clip-Cov` and `KL-Cov`, which clip and apply KL penalty to tokens with high covariances respectively. Experiments show that these methods encourage exploration, thus helping policy escape entropy collapse and achieve better downstream performance.

## 1 INTRODUCTION

Applied to recent reasoning-centric large language models (LLMs), reinforcement learning (RL) escapes narrow, task-specific confines: the models' sweeping generalization introduces a new axis that vastly enlarges the exploratory landscape. This shift has yielded impressive reasoning gains (OpenAI, 2024a; DeepSeek-AI et al., 2025), yet the dilemma persists—scaling training compute for *learning from experience* (reinforcement learning) (Silver & Sutton, 2025) rather than *imitation learning* (pre-training and finetuning) remains non-trivial. Among the challenges emerges a major obstacle, the diminishment of *policy entropy*.

The core challenge in RL is the exploitation-exploration trade-off (Sutton, 1988), balancing the reuse of proven strategies against the search for novel ones. For exploration, a key concept quantifying the exploratory potential is *policy entropy*, which measures the uncertainty in the policy's action selection process. In RL literature, the ability to mitigate the decline of policy entropy is regarded as essential to most algorithms (Williams & Peng, 1991; Williams, 1992; Eysenbach & Levine, 2021), and policy entropy has been intensively steered and actively controlled via regularization (Ziebart et al., 2008; Schulman et al., 2017b; Haarnoja et al., 2018).

For LLMs, while the typical behavior of policy entropy remains largely understudied (Yu et al., 2025; He et al., 2025), we find an intriguing and consistent pattern from broad experiments: Policy entropy sharply declines to near 0 in a few training steps, demonstrating that the policy becomes extremely

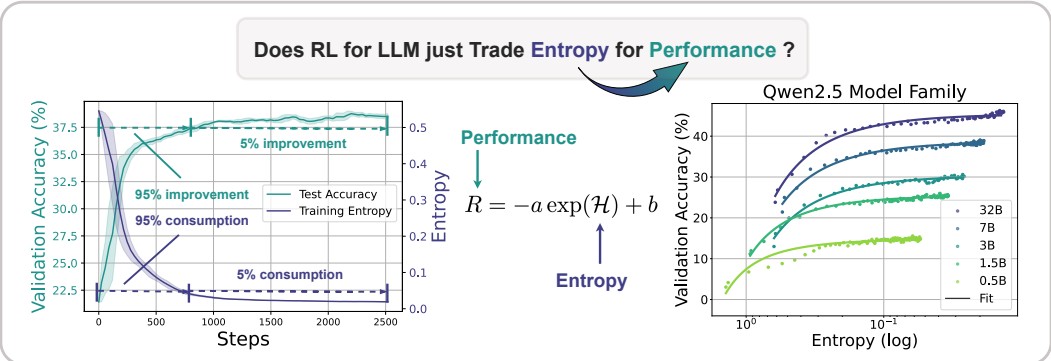

Figure 1: *Left:* Entropy collapse and performance saturation. Over $95\%$ entropy drop/performance gains take place at the early stage of RL training. The model then reaches a plateau with little improvement. *Right:* The predictable relationship between validation performance and policy entropy. Without intervention, the policy "trades" entropy for performance exponentially, showing clear ceilings that hinder further policy enhancement.

certain. Consequently, the inability to explore new paths leads to a performance plateau, where the validation performance also struggles to improve at the same time. Quantitatively, we further reveal that, without entropy intervention like entropy loss or KL regularization, **the downstream performance is fully predictable from policy entropy, and the fitted curve is a simple exponential function $R = -a \exp \mathcal{H} + b$, as shown in Figure 1.** Basically, the policy is trading uncertainty (entropy) for rewards (Yue et al., 2025) *in a predictable manner*.

This empirical law results in two critical corollaries: (1) Like Scaling Laws (Kaplan et al., 2020; Hoffmann et al., 2022), the exploitation-exploration curve is predetermined given the policy model and training data. This allows us to predict policy performance at the early stage of RL and predict the performance of large models given small models (OpenAI, 2024b) (Sec. 2.4). (2) More importantly, this equation indicates that the upper bound of the policy performance is also deterministic with the exhaustion of policy entropy ($\mathcal{H} = 0, R = -a + b$), so the return of scaling training compute for RL could be marginal. What's worse, naively applying entropy regularization methods are proven ineffective (App. E). **In short, scalable RL calls for breaking the entropy bottleneck.**

Solving this issue requires principled understandings of the mechanisms behind this observation, i.e., *why policy entropy decreases monotonically?* To this end, we analyze the dynamics of policy entropy both theoretically and empirically. Our key findings highlight that, for softmax policy like LLMs, the entropy change between two consecutive steps is proportional to the covariance of the log-probability and corresponding logit change for an action (Liu, 2025). Furthermore, under Policy Gradient (Williams, 1992)-like and Natural Policy Gradient (Kakade, 2001)-like algorithms, the logit difference is proportional to the action advantage. Intuitively, an action with high advantage and high probability would reduce policy entropy, while a rare action with a high advantage would increase entropy. This theoretical conclusion is validated by experimental results. At the early stage, the policy demonstrates high covariance on training data, implicating the policy's confidence is well-calibrated (Kadavath et al., 2022), thus can safely exploit trajectories with high confidence, strengthening belief and minimize entropy (Zuo et al., 2025; Zhang et al., 2025; Agarwal et al., 2025). As training progresses, the covariance gradually declines but still maintains positive, continually dragging policy entropy even lower.

The analysis of entropy dynamics demonstrates that, the high covariance is detrimental to scalable RL, which provides us guidelines about uplifting policy entropy—limit the step sizes of high-covariance tokens. We thereby motivate to design two corresponding strategies aiming at entropy control, namely `Clip-Cov` and `KL-Cov`, to replace the clip and PPO-KL methods in surrogate loss (Schulman et al., 2017b). `Clip-Cov` randomly selects a small portion of tokens with positive covariances and detach their gradients. `KL-Cov`, on the other hand, applies KL penalty on tokens with the largest covariances. Experiment results show that, we can actively control policy entropy by tuning threshold parameters. Consequently, the policy model escapes the low entropy trap and achieves better performance on mathematical reasoning.

## 2 THE PREDICTABLE "COLLAPSE" OF POLICY ENTROPY

**TAKEAWAY**

Without intervention, e.g., entropy or KL regularization, policy entropy is ***traded for reward predictably*** during RL. The empirical quantitative relationship between validation reward $R$ and policy entropy $\mathcal{H}$ can be expressed as $R = -a \exp(\mathcal{H}) + b$.

In this section, we manage to answer the research question: *What is the typical behavior of policy entropy during RL for LLMs?* Through extensive experiments, we observe a consistent "entropy collapse" phenomenon (Sec. 2.3), and further extend it to an empirically predictable relation between policy entropy and validation performance (Sec. 2.4).

### 2.1 PRELIMINARIES

We consider tuning LLMs with RL on verifiable tasks, such as math and coding, to avoid reward hacking. Given an input prompt $\boldsymbol{x}$, an LLM $\pi_\theta$ autoregressively generates an output sequence $\boldsymbol{y}$, which consists of $T$ tokens $\{y_1, \cdots, y_t, \cdots, y_T\}$. We use RL to optimize the LLM policy to maximize the cumulative rewards $r$ received from the verifier:

$$\max_\theta J(\theta) := \mathbb{E}_{\boldsymbol{x} \sim \mathcal{D}, \boldsymbol{y} \sim \pi_\theta(\boldsymbol{x})} [r(\boldsymbol{y})] \tag{1}$$

where $\mathcal{D}$ is the training distribution.

To optimize the objective function, it is a common practice to use the Policy Gradient algorithm (Williams, 1992) for gradient estimation:

$$\nabla_\theta J(\theta) = \mathbb{E}_{\boldsymbol{x} \sim \mathcal{D}, \boldsymbol{y} \sim \pi_\theta(\boldsymbol{x})} \left[ \sum_{t=0}^{T} \nabla_\theta \log \pi_\theta(y_t | \boldsymbol{y}_{<t}) A_t \right]. \tag{2}$$

Here $A_t$ is the advantage of current action and is implemented differently across RL algorithms. If we only have rewards for the full trajectory, the vanilla REINFORCE algorithm (Williams, 1992) directly defines $A_t = r(\boldsymbol{y})$. To reduce variance, GRPO (Shao et al., 2024) and RLOO (Kool et al., 2019; Ahmadian et al., 2024) further incorporates group-wise normalization. For example, GRPO samples $K$ responses for each prompt and estimates the advantage as follows:

$$A_t = \frac{r(\boldsymbol{y}) - \text{mean}(r(\boldsymbol{y}^{1:K}))}{\text{std}(r(\boldsymbol{y}^{1:K}))}. \tag{3}$$

To handle off-policy data and constrain the policy update size, PPO (Schulman et al., 2017b) proposed to optimize a surrogate loss.

$$L(\theta) = \mathbb{E}_t \left[ \min\left( \frac{\pi_\theta(y_t|\boldsymbol{y}_{<t})}{\pi_{\theta_{old}}(y_t|\boldsymbol{y}_{<t})} A_t, \text{clip}\left( \frac{\pi_\theta(y_t|\boldsymbol{y}_{<t})}{\pi_{\theta_{old}}(y_t|\boldsymbol{y}_{<t})}, 1-\epsilon, 1+\epsilon \right) A_t \right) \right] \tag{4}$$

**Policy Entropy.** Policy entropy quantifies the predictability or randomness inherent in the actions selected by an agent. Given policy model $\pi_\theta$, training dataset $\mathcal{D}$, we measure the average token-level entropy of the policy model on training data, which is defined as follows:

$$\mathcal{H}(\pi_\theta, \mathcal{D}) = -\mathbb{E}_{\mathcal{D}, \pi_\theta} [\log \pi_\theta(y_t|\boldsymbol{y}_{<t})] = -\frac{1}{|\mathcal{D}|} \sum_{x \in \mathcal{D}} \frac{1}{|\boldsymbol{y}|} \sum_{t=1}^{|\boldsymbol{y}|} \mathbb{E}_{y_t \sim \pi_\theta} [\log \pi_\theta(y_t|\boldsymbol{y}_{<t}, x)] \tag{5}$$

Such entropy quantifies the uncertainty level of the policy on current prompts and is widely adopted in maximum entropy RL as a regularization term (Ziebart et al., 2008; Haarnoja et al., 2017; 2018). In practice, we calculate the entropy for each batch sampled from the training dataset.

### 2.2 EXPERIMENT SETTINGS

We adopt a unified protocol covering 4 model families and 11 base models (0.5-32B parameters), verifiable task domains of math and coding evaluated on 8 public benchmarks, and 4 RL algorithms.

We start RL from the base models following the "Zero" setting (DeepSeek-AI et al., 2025) with the veRL framework (Sheng et al., 2024). For RL algorithms, we employ GRPO (Shao et al., 2024), REINFORCE++ (Hu, 2025), and PRIME (Cui et al., 2025). The details of the used models, datasets, and hyperparameters can be found in Appendix B.1.

### 2.3 A First Glance: Entropy Collapse and Performance Saturation

Across all experiments, we observe a consistent pattern: policy entropy drops sharply at the beginning of training, declining monotonically toward zero. Meanwhile, the policy's validation performance presents an inverse trend, then plateaus.

Figure 2 illustrates the average normalized entropy consumption/performance gain in percentage throughout 2400-gradient step RL runs with 11 different models. We can see that 73% of the entropy consumption and 76% of the performance gain occurred in just the first 200 gradient steps (1/12 of training), and the first 800 (1/3) steps account for over 93% performance gains together with 94% entropy losses. This means that over 2/3 of the training steps yielded marginal returns.

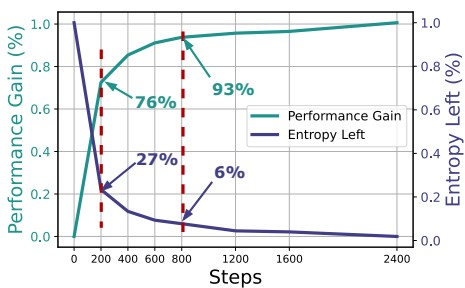

Figure 2: Avg. entropy consumption and performance gain (%) in 11 RL runs with different models.

### 2.4 Fitting the Curves between Entropy and Performance

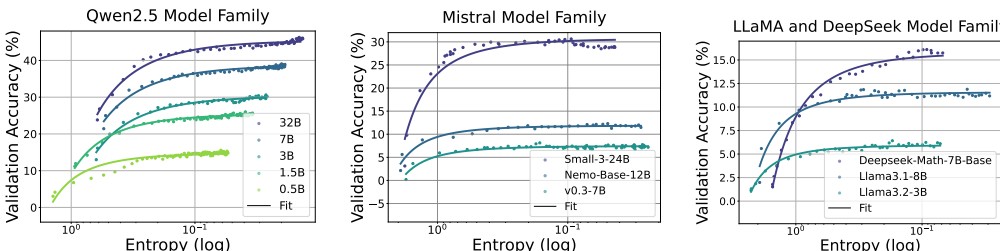

Figure 3: Fitting curves between policy entropy and validation performance on math task.

Motivated by the observed entropy collapse phenomenon, we conduct a more detailed quantitative analysis. Through extensive experiments, we find the downstream performance (accuracy) and entropy can be fitted in the exponential function:

$$R = -a \exp(\mathcal{H}) + b, \tag{6}$$

where $R$ represents the validation performance and $\mathcal{H}$ is entropy. The fitting results of different model families with GRPO are presented in Figure 3 and 7. It is worth noting that, the fitted curves precisely describe the performance-entropy relationships over all conducted experiments, with models spanning all kinds of sizes, families, and different tasks. Only 2 coefficients are needed for fitting the curve of over 200 data points, showing a high degree of regularity. Detailed results on more tasks, datasets, and algorithms can be found in App. B.2. We further analyzed the predictability in App. B.3.

### 2.5 Discussion

**The Predictability.** To now, we have established predictability between (1) policy performance and entropy; (2) coefficients in (1) and model sizes. Such predictability reminds us of Scaling Laws for language models (Kaplan et al., 2020; Hoffmann et al., 2022) and RLHF (Gao et al., 2022). It seems that, RL with LLMs keeps trading entropy for reward throughout training. However, other works that adopt different policy models (Luo et al., 2025) or use off-policy data (Yan et al., 2025) observed

distinct entropy patterns. Therefore, this predictability is not arguably universal, and we call for a more in-depth analysis of the entropy behavior under different conditions.

**The Ceiling.** There is an intensive discussion questioning whether RL merely elicits the latent behaviors that were already learned in pre-training, thus cannot break the ceiling of the base model (Yue et al., 2025). Our results conditionally support this claim that, if policy entropy diminishes, the ceiling not only exists, but also can be predicted. However, we argue that it is not the intrinsic limitation of RL that sets up the ceiling, but the entropy mechanism of LLMs leads to the result.

## 3 DYNAMICS ANALYSIS OF POLICY ENTROPY

> **TAKEAWAY**
>
> For softmax policy including LLMs, the change of policy entropy is determined by the **covariance** between the log-probability and the change in logits of actions. For Policy Gradient and Natural Policy Gradient, the change in logits is proportional to the action advantage, meaning that a high covariance leads to a quick decrease of policy entropy, as observed in RL for LLM reasoning.

We have unveiled that the entropy collapse issue will greatly obstacle RL scaling for LLM reasoning. To solve it, we need a more principled understanding of the *dynamics of policy entropy*, *i.e.*, when will entropy decrease and when will entropy increase. In this section, we focus on the entropy dynamics, especially the step-wise entropy difference $\mathcal{H}(\pi_\theta^{k+1}) - \mathcal{H}(\pi_\theta^k)$.

### 3.1 ENTROPY DYNAMICS OF SOFTMAX POLICY

For step $k$, we try to calculate the entropy difference before and after one step parameter update, *i.e.*, $\mathcal{H}(\pi_\theta^{k+1})$ and $\mathcal{H}(\pi_\theta^k)$. For this, we first consider an intrinsic property of LLMs that they are softmax policies, which means the policies are parameterized by $\pi_\theta(a|s) = \frac{\exp(z_{s,a})}{\sum_{a' \in \mathcal{A}} \exp(z_{s,a'})}$. Here $s \sim d_{\pi_\theta}$ and $a \sim \pi_\theta^k(\cdot|s)$ represent state and action, $z_{s,a}$ is the output logit of action $a$ given state $s$. For any softmax policy, we have the following Lemma:

**Lemma 1 (Entropy difference of softmax policy)** *(Proof in Appendix C.2, adapted from Liu (2025)) Assume that policy $\pi_\theta$ is a tabular softmax policy, where each state-action pair $(s, a)$ is associated with an individual logit parameter $z_{s,a} = \theta_{s,a}$, the difference of policy entropy given state $s$ between two consecutive steps under first-order approximation satisfies*

$$\mathcal{H}(\pi_\theta^{k+1}) - \mathcal{H}(\pi_\theta^k) \approx \mathbb{E}_{s \sim d_{\pi_\theta}} \left[ \mathcal{H}(\pi_\theta^{k+1}|s) - \mathcal{H}(\pi_\theta^k|s) \right] \approx \mathbb{E}_{s \sim d_{\pi_\theta}} \left[ -Cov_{a \sim \pi_\theta^k(\cdot|s)} \left( \log \pi_\theta^k(a|s), \, z_{s,a}^{k+1} - z_{s,a}^k \right) \right]$$

Here $z_{s,a}^{k+1} - z_{s,a}^k$ is the change in the output logits between step $k$ and step $k + 1$. This Lemma indicates that, the change of policy entropy approximately equals the negative covariance between log-probability of the action and the change of logits. That is to say, when an action $a$ receives a high probability from the policy before updating, and its corresponding logit is also increasing after updating, then it will decrease the policy entropy.

### 3.2 ENTROPY DYNAMICS UNDER POLICY GRADIENT / NATURAL POLICY GRADIENT

From Lemma 1, the step-wise difference of output logits $z_{s,a}^{k+1} - z_{s,a}^k$ contributes to change of entropy, which is related with the specific training algorithm in use. Here, we further derive the logits change under Policy Gradient (Williams, 1992) and Natural Policy Gradient (Kakade, 2001) algorithms.

Assuming that we are updating the actor policy via Policy Gradient, then $z_{s,a}^{k+1} - z_{s,a}^k = -\eta \cdot \nabla_z J(\theta)$, where $J(\theta)$ denotes the objective function and $\eta$ denote the learning rate. $\nabla_z J(\theta)$ is calculated with Eq. 2, we have the following proposition:

**Proposition 1 (Difference of policy logits in vanilla policy gradient)** *(Proof in Appendix C.3) Let the actor policy $\pi_\theta$ be a tabular softmax policy and updated using Eq. 2 via gradient backtracking with learning rate $\eta$, the difference of $z_{s,a}$ between two consecutive steps satisfies*

$$z_{s,a}^{k+1} - z_{s,a}^k = \eta \, \pi_\theta(a \mid s) \, A(s, a)$$

Applying Proposition 1 to Lemma 1, we can further describe entropy change with:

**Theorem 1 (Entropy change under policy gradient)** *Let the actor policy $\pi_\theta$ be a tabular softmax policy, and $\pi_\theta$ be updated via vanilla policy gradient, the difference of policy entropy given state $s$ between two consecutive steps satisfies*

$$\mathcal{H}(\pi_\theta^{k+1}|s) - \mathcal{H}(\pi_\theta^k|s) \approx -\eta \cdot Cov_{a \sim \pi_\theta^k(\cdot|s)} \left( \log \pi_\theta^k(a|s) , \pi_\theta^k(a|s) \cdot A(s,a) \right)$$

Theorem 1 reveals how policy entropy changes under the policy gradient method. Intuitively, an action $a$ receives both high/low probability and high/low advantage would lower the entropy, and vice versa. Liu (2025) conducted derivation for Natural Policy Gradient.

**Theorem 2 (Entropy change under natural policy gradient)** *(Proof in Appendix C.4) Let the actor policy $\pi_\theta$ be a tabular softmax policy, and $\pi_\theta$ is updated via natural policy gradient (Kakade, 2001), the difference of policy entropy given state $s$ between two consecutive steps satisfies*

$$\mathcal{H}(\pi_\theta^{k+1}|s) - \mathcal{H}(\pi_\theta^k|s) \approx -\eta \cdot Cov_{a \sim \pi_\theta^k(\cdot|s)} \left( \log \pi_\theta^k(a|s) , A(s,a) \right)$$

**Conclusion.** From Theorem 1 and Theorem 2, we obtain the intuitive insight that, in principle, a strong positive correlation between the action probability $P(a)$ under the current policy and the corresponding advantage value $A(a)$, on average, leads to a decrease in policy entropy. Conversely, a negative correlation tends to increase the entropy. This deeper understanding of the dynamics of policy entropy provides a theoretical foundation for designing practical strategies for entropy control.

### 3.3 Empirical Verification

In this section, we conduct experiments to validate the theoretical conclusion, specifically, Theorem 1.

**Settings.** We apply GRPO with policy gradient, *i.e.* on-policy learning without PPO surrogate loss, on Qwen2.5-7B. In this context, we adopt the bandit setting where the prompt $\boldsymbol{x}$ is the state, and whole response $\boldsymbol{y}$ is the action. Then the covariance term becomes:

$$Cov_{a \sim \pi_\theta(\cdot|s)} \left( \log \pi_\theta(a \mid s), \pi_\theta(a \mid s) \cdot A(s,a) \right) = Cov_{\boldsymbol{y} \sim \pi_\theta(\cdot|\boldsymbol{x})} \left( \log \pi_\theta(\boldsymbol{y} \mid \boldsymbol{x}), \pi_\theta(\boldsymbol{y} \mid x) \cdot A(\boldsymbol{y}, \boldsymbol{x}) \right) \quad (7)$$

During training, we calculate the group-wise covariance for each prompt, and average across a batch of prompts. We further normalize the log-prob by the length of the response.

**Experiment Results.** We record two key metrics, $Cov(\cdot)$ and $\mathcal{H}(\pi_\theta)$, across training and analyse their relationship. According to Theorem 1, we have $-d(\mathcal{H}) \propto Cov(\cdot)$. As shown in Figure 4, the empirical curves of $-d(\mathcal{H})$ and $Cov(\cdot)$ exhibit highly similar dynamics. Early in training, entropy $\mathcal{H}$ decreases rapidly, accompanied by a relatively large positive $Cov(\cdot)$. As training progresses, entropy decay slows and $Cov(\cdot)$ stabilizes, reflecting convergence of the policy. Notably, $Cov(\cdot)$ remains positive throughout training, leading to a persistent entropy decrease and finally collapse.

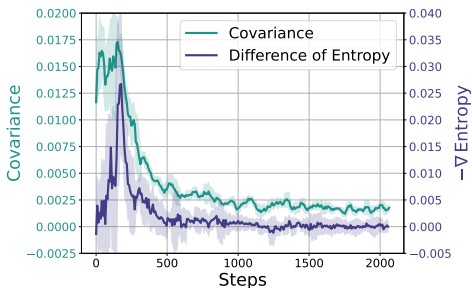

Figure 4: Dynamics of policy entropy difference and covariance during GRPO training. They show similar trends, as predicted by theory.

## 4 Entropy Control by Covariance Regularization

> **TAKEAWAY**
>
> We can control policy entropy by **restricting the update of tokens with high covariances**, e.g., clipping (`Clip-Cov`) or applying KL penalty (`KL-Cov`). These simple techniques prevent policy from entropy collapse thus promoting exploration.

The analysis of entropy dynamics gives us guidelines for entropy control, regularizing the update step size of high-covariance actions. In this section, we introduce two simple yet effective techniques, `KL-Cov` and `Clip-Cov`, that control entropy precisely and achieve better downstream performance.

### 4.1 Suppressing Tokens with High Covariances

To get the entropy controlled, we conduct experiments on the common approaches in the RL literature, however, results show that those approaches struggles to solve the entropy bottleneck of LLMs (See Appendix E). The unsuccessful attempt to incorporate entropy regularization into the policy loss drives us to seek a more fundamental approach to control entropy. As previously elaborated, the policy entropy dynamic is closely connected with the covariance between action probability and advantage. Meanwhile, as shown in Table 1, a small portion of tokens exhibit extremely high covariance, far exceeding the average. That is saying that these outlier tokens take a dominant part in triggering the entropy collapse. To mitigate their

Table 1: Covariance distribution of Qwen2.5-7B in training step 1.

| Group | Mean Value |
|---|---|
| Top 0.02% | 5.654 |
| Top 0.2% | 3.112 |
| Top 2% | 1.385 |
| Top 20% | 0.351 |
| All | 0.003 |

adverse effect, we aim to impose constraints on their contribution to the policy loss. In RL literature, two variants of PPO employ either clipping or KL penalty to constrain the policy updates (Schulman et al., 2017b), preventing overly aggressive changes. Drawing inspiration from these approaches, we propose two simple but effective covariance-aware methods `Clip-Cov` and `KL-Cov`.

Natural policy gradient is rarely used in post-training of LLMs for its time-consuming second-order optimization. But its introduction of target function with KL distance as constraint shares similarity with TRPO (Schulman et al., 2015) and PPO. Thus, we apply Theorem 2 into RL training.

Supposing a batch of $N$ rollout tokens, $\pi_\theta(y_i)$ denotes the output probability of the policy model for token $y_i$ given its corresponding prefix. According to Theorem 2, we firstly define token-wise centered cross-product between log probability and advantage as:

$$Cov(y_i) = (\log \pi_\theta(y_i) - \frac{1}{N}\sum_{j=1}^{N} \log \pi_\theta(y_j)) \cdot (A(y_i) - \frac{1}{N}\sum_{j=1}^{N} A(y_j)) \tag{8}$$

The $Cov$ is the covariance of each token in $N$. Its expectation is the covariance in Theorem 2.

**Clip-Cov.** In the `Clip-Cov` strategy, we clip a small fraction of high-covariance tokens out from policy gradient updates as follows. With Eq. 8 calculated, we randomly select $k \cdot N$ of high-covariance tokens according to the covariance value:

$$I_{\text{clip}} = I \sim \text{Uniform}\left(i \mid Cov(y_i) \in [\omega_{\text{low}}, \omega_{\text{high}}]\}, \lfloor k \cdot N \rfloor\right) \tag{9}$$

Where $I$ is short for index, $k$ denotes the clip ratio. $\omega_{\text{low}}, \omega_{\text{high}}$ are two predefined bounds for covariance, respectively. Both are set much higher than the average covariance ($>500\times$). Finally, tokens with the chosen indexes will be detached from the policy gradient, which is:

$$L_{\text{Clip-Cov}}(\theta) = \begin{cases} \mathbb{E}_t\left[\frac{\pi_\theta(y_t|\boldsymbol{y}_{<t})}{\pi_{\theta_{\text{old}}}(y_t|\boldsymbol{y}_{<t})}A_t\right], & t \notin I_{\text{clip}} \\ 0, & t \in I_{\text{clip}} \end{cases} \tag{10}$$

where the $t$ is the $t$-th token in one rollout response and each $t$ uniquely corresponds to a index $i$.

**KL-Cov.** The `KL-Cov` strategy is simpler. Specifically, similar to `Clip-Cov`, we first compute the covariance as in Eq. 8. Then, we rank and select tokens within the top-$k$ proportion of covariance:

$$I_{\text{KL}} = \{i \mid \text{Rank}(Cov(y_i)) \leq k \cdot N\}, \tag{11}$$

The $k$ here denotes the proportion of tokens that will be subjected to the KL penalty and $k \ll 1$. At last, we impose the KL penalty (KL divergence between the current policy and the rollout policy) on the selected tokens, the policy loss is computed as:

$$L_{\text{KL-Cov}}(\theta) = \begin{cases} \mathbb{E}_t\left[\frac{\pi_\theta(y_t|\boldsymbol{y}_{<t})}{\pi_{\theta_{\text{old}}}(y_t|\boldsymbol{y}_{<t})}A_t\right], & t \notin I_{\text{KL}} \\ \mathbb{E}_t\left[\frac{\pi_\theta(y_t|\boldsymbol{y}_{<t})}{\pi_{\theta_{\text{old}}}(y_t|\boldsymbol{y}_{<t})}A_t - \beta\mathbb{D}_{\text{KL}}(\pi_{\theta_{\text{old}}}(y_t|\boldsymbol{y}_{<t})||\pi_\theta(y_t|\boldsymbol{y}_{<t}))\right], & t \in I_{\text{KL}} \end{cases} \tag{12}$$

Where $\beta$ is the coef. to control the weight for KL penalty. Pseudo-code is presented in Algorithm 1.

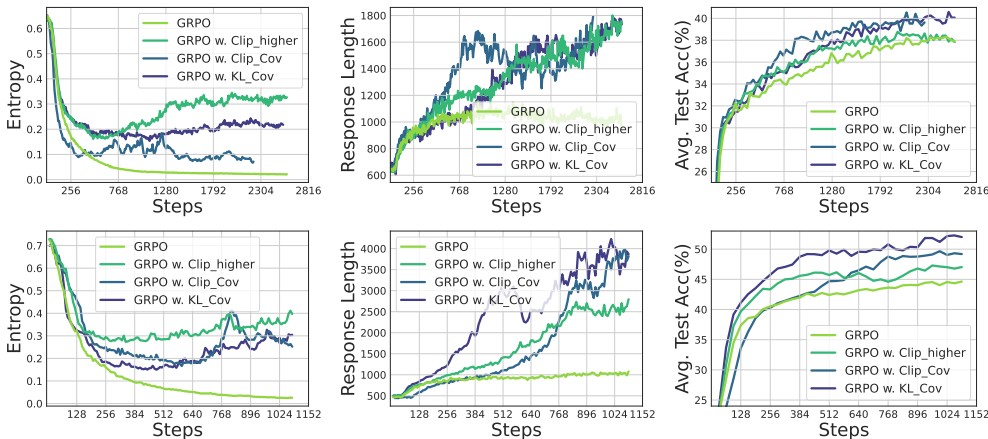

Figure 5: Training Qwen2.5-7B (*Top*) / Qwen2.5-32B (*bottom*) with GRPO with/without our methods. *Left:* Entropy dynamics. Our methods uplift policy entropy from collapse, enabling sustained exploration. *Middle:* Our methods also incentivize longer responses compared with vanilla GRPO. *Right:* Our methods consistently outperform baselines on testsets.

Table 2: Detailed results of GRPO, GRPO with clip-higher technique and our methods. For AIME and AMC, the results are avg.@32. **Bold** denotes the best results.

| Method | AIME24 | AIME25 | AMC | MATH-500 | OMNI-MATH | OlympiadBench | Minerva | Avg. |
|---|---|---|---|---|---|---|---|---|
| *Llama3.1-8B* | | | | | | | | |
| GRPO | 0.3 | 0.4 | 7.3 | **25.8** | 7.2 | 5.6 | 9.2 | 6.8 |
| w. Clip-higher | 0.0 | 0.0 | 8.5 | 23.0 | 7.3 | 4.7 | 12.1 | 6.9 |
| w. **Clip-Cov** | **0.4** | 0.3 | 8.9 | 23.4 | **8.6** | **7.3** | 12.5 | **7.8** |
| w. **KL-Cov** | 0.4 | **0.7** | **9.1** | 23.0 | 7.3 | 4.1 | **13.2** | 7.2 |
| *Qwen2.5-7B* | | | | | | | | |
| GRPO | 21.2 | 9.6 | 58.7 | 78.8 | 27.9 | 40.7 | 36.7 | 38.6 |
| w. Clip-higher | 18.1 | 11.5 | 56.6 | 79.2 | 29.8 | 43.3 | 40.4 | 38.8 |
| w. **CLIP-Cov** | 22.1 | **15.8** | 58.2 | 80.4 | **30.5** | **44.1** | **41.1** | 40.4 |
| w. **KL-Cov** | **22.6** | 12.9 | **61.4** | **80.8** | 29.1 | 42.6 | 38.2 | **40.6** |
| *Qwen2.5-32B* | | | | | | | | |
| GRPO | 21.8 | 16.2 | 69.7 | 84.2 | 35.2 | 43.6 | 45.5 | 45.8 |
| w. Clip-higher | 35.6 | 22.3 | 69.5 | 77.2 | 35.1 | 42.5 | 43.0 | 47.2 |
| w. **CLIP-Cov** | 32.3 | 22.7 | 67.2 | **87.0** | **42.0** | **57.2** | 46.0 | 50.3 |
| w. **KL-Cov** | **36.8** | **30.8** | **74.5** | 84.6 | 39.1 | 49.0 | **46.3** | **52.2** |
| *Qwen3-8B* | | | | | | | | |
| GRPO | 31.7 | 22.9 | 65.3 | 87.6 | 39.5 | 54.6 | 45.2 | 48.7 |
| w. Clip-higher | 33.8 | **27.1** | **73.1** | 89.0 | 39.9 | 52.2 | 46.3 | 51.8 |
| w. **Clip-Cov** | 31.9 | 24.5 | 71.8 | **90.0** | **43.7** | **60.4** | 46.7 | 52.8 |
| w. **KL-Cov** | **36.7** | 26.5 | 72.4 | 87.8 | **43.7** | 58.4 | **47.4** | **53.5** |

## 4.2 EXPERIMENTS

**Settings.** Because of the capability differences across base models, we train Llama3.1-8B with GSM8K, while other models are trained using the DAPO-MATH dataset. For baselines, we compare the original GRPO, and GRPO with Clip-higher, which tunes the upper threshold $\epsilon$ in PPO loss to 0.28 (Yu et al., 2025). More details about the training hyperparameters can be found in Appendix F.2.

**Results and Analysis.** We present the experimental results in Table 2, one can see that our two approaches both achieve non-trivial improvements across all benchmarks. Compared to GRPO, our method outperforms it by 2.0% on average for the 7B model and by 6.4% for the 32B model.

As shown in Figure 5, our method is able to maintain a considerably higher level of entropy throughout training. For example, when the baseline's entropy reaches a plateau and can no longer be consumed, the KL-Cov method still sustains an entropy level over 10× higher. Meanwhile, the response length of the policy model steadily increases, and its performance on the test set consistently surpasses that of the baseline. This indicates that our policy model is able to explore more "freely" during

training. Compared to the clip-higher technique, although it can also increase entropy and lead to performance improvement in the early stage of training, it gradually becomes unstable, with performance saturating and declining. In contrast, our method obtains more stable entropy curves throughout training, ultimately achieving non-trivial improvements over the baselines.

Moreover, we observe that our method yields more substantial gains on Qwen2.5-32B. Specifically, we achieve improvements of **15.0%** and **14.6%** compared to GRPO on the most challenging benchmarks, AIME24 and AIME25, respectively. We infer that this is because the 32B model possesses greater potential from pretraining compared to the 7B model. Once the "exploration curse" caused by entropy collapse is lifted, the larger model is able to explore more diverse and higher-quality policies.

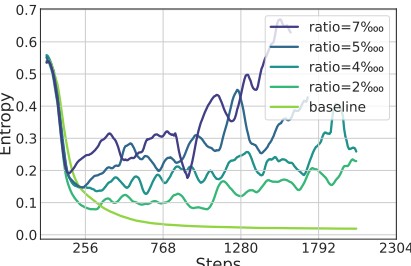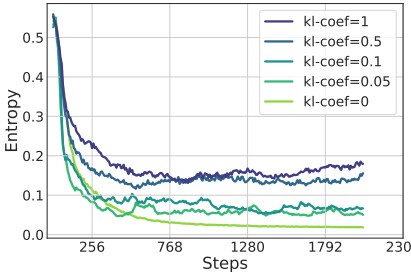

Figure 6: Differences in entropy dynamics of Qwen2.5-7B under varying KL coefficients and Clip ratios, evaluated `Clip-Cov` (*left*) and `KL-Cov` (*right*) settings, respectively.

### 4.3 GET POLICY ENTROPY CONTROLLED

We also evaluate the capability of our methods in controlling policy entropy as shown in Figure 6. For `Clip-Cov`, the level of policy entropy can be adjusted by tuning the ratio of clipped samples, where more clipped samples result in higher entropy. For `KL-Cov`, we can modulate the entropy by controlling the KL coefficient $\beta$, *i.e.*, the weight of the KL penalty. Specifically, a larger coefficient brings higher entropy. Comparing them, `KL-Cov` reaches stabler entropy curves than `Clip-Cov`, which might be preferable for stabilizing the training process.

### 4.4 DISCUSSION

**Connection with Clip-higher.** Our main baseline, clip-higher (Yu et al., 2025), can also incentivize higher policy entropy. In fact, this technique shares similar functionality with our methods. By raising the upper threshold of the importance sampling ratio, clip-higher includes more low-probability tokens for policy updates. Also, the upper threshold only affects the tokens with positive advantages, which means clip-higher is actually adding more low-covariance (low probability, high advantage, with average covariance of ∼-0.03) tokens in gradient calculation. We take a step further by directly using the covariance as the threshold, thus controlling the entropy more precisely.

**The Philosophy of Entropy Control.** In experiments, we find that the policy entropy is sensitive to hyperparameter settings. Specifically, our methods only interfere with a very small fraction of tokens ($10^{-4}$ to $10^{-3}$), yet totally change the entropy curve. This means several "pivotal" tokens are crucial for the entropy of LLMs. However, we don't observe a relationship between the intervened entropy and model performance. It still remains open whether there exists an optimal entropy value to balance the exploration and training stability.

## 5 CONCLUSION

In this study, we try to address the challenge of policy entropy collapse in reinforcement learning for large language model reasoning. We empirically demonstrate that performance gains are often achieved by sacrificing exploratory capacity, which in turn imposes a foreseeable limit on improvement. To gain a deeper understanding, we conduct a theoretical investigation into entropy dynamics and introduce two simple regularization techniques, `Clip-Cov` and `KL-Cov`, to directly manage high-covariance tokens and thereby counteract entropy collapse. Looking further, RL has been identified as the next scaling axis after pre-training. However, scaling computing for RL requires more than entropy minimization. We hope this research could provide valuable insights into the role of entropy, fostering RL to reach a higher level of intelligence.

## REPRODUCIBILITY STATEMENT

We have provided sufficient details to for reproduction, including algorithm pseudocode in Algorithm 1, experiment configurations and hyperparameters in Section 2, Section 4 and Appendix. We have uploaded our code in Supplementary Material.

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

## A  RELATED WORK

**Policy Entropy in Reinforcement Learning.** Stemmed in information theory, entropy provides a principled mechanism to manage the exploitation-exploration tradeoff. Entropy-regularized reinforcement learning, also referred as maximum entropy RL (Ziebart et al., 2008; Toussaint, 2009), adopts a regularization term in reward to encourage high-entropy actions. This regularization term was widely-inherited in RL algorithms (Mnih et al., 2015; 2016; Schulman et al., 2017a;b; Haarnoja et al., 2017; 2018), and is viewed as a necessity. On the other hand, in RL for LLMs, there exist different opinions on whether entropy regularization should be preserved (Ouyang et al., 2022; Shao et al., 2024; Hu et al., 2025; He et al., 2025). Our experiments indicate that, it is necessary to control entropy, but we can design better objectives than entropy loss.

**Predictability of Reinforcement Learning for Reasoning Language Models.** The first part of this work reveals the predictability of RL for LLM reasoning. The development of LLMs is largely guided by the neural scaling laws, which bridge model performances with computational budgets, model sizes, and the amount of training data (Hestness et al., 2017; Kaplan et al., 2020; Hoffmann et al., 2022). With scaling experiments on smaller models, the loss and task performance of larger models could be accurately predicted. In RL, Hilton et al. (2023); Rybkin et al. (2025) studied the scaling behavior of policy performances versus computing on non-LLM models, but the predictability of RL for LLMs has yet to be investigated. Gao et al. (2022) proposed to predict reward scores from KL divergence in RL on LLMs, which was used for modeling overoptimization effect of a proxy reward model. This work aligns with our conclusion considering that, 1) the verifiable reward eliminates the gap between the proxy reward model and ground truth; 2) the similarity between KL divergence and policy entropy.

**Reinforcement Learning for LLMs.** Reinforcement learning has emerged as a major approach for LLM post-training (Ouyang et al., 2022; Meta, 2024; Team et al., 2023; Qwen et al., 2025; Jiang et al., 2023). Recent works have achieved further breakthrough on enhancing the reasoning capability of LLMs using RL with verifiable rewards (OpenAI, 2024a; Lambert et al., 2024; DeepSeek-AI et al., 2025; Team et al., 2025), drawing great attention in research community (Cui et al., 2025; Liu et al., 2025; Hu et al., 2025; He et al., 2025). However, there still lacks systematic study on the underlying mechanisms of RL for LLMs, which constitutes the primary goal of our work.

## B  DETAILED RESULTS AND DISCUSSION OF THE FITTING EXPERIMENTS

### B.1  EXPERIMENT SETTINGS

**Models.** The models adopted in our experiments span 4 model families and 11 widely used open-source base models. Specifically, these consist of the Qwen2.5 family (Qwen2.5-0.5B, 1.5B, 3B, 7B, 32B) (Qwen et al., 2025), the Mistral family (Mistral-7B-v0.3 (Jiang et al., 2023), Mistral-Nemo-Base-2407 (MistralAI-NeMo), Mistral-Small-3.1-24B-Base-2501 (MistralAI-Small-3)), the LLaMA family (LLaMA3.2-3B (Meta-Llama-3.2), LLaMA3.1-8B (Meta, 2024)), and DeepSeek-Math-7B-Base (Shao et al., 2024)).

**Tasks and Datasets.** We primarily focus on math and coding problems with verifiable rewards. Due to inherent differences in the initial reasoning abilities between model families, we train models using data of different difficulty levels to stabilize the RL process. Meanwhile, we use the same data during downstream performance evaluation to maintain consistency. For math tasks, the evaluation datasets include MATH500 (Hendrycks et al., 2021), AIME 2024 (Li et al., 2024), AMC (Li et al., 2024), OlympiadBench (He et al., 2024), and OMNI-MATH (Gao et al., 2024). For code tasks, we split the testset of Eurus-2-RL-Code (Cui et al., 2025) and KodCode (Xu et al., 2025).

Specifically, Due to inherent differences in the initial reasoning abilities between model families, we train models using data of different difficulty levels to stabilize the RL process Specifically, for math tasks, we train the Qwen family and Mistral-24B model using Eurus-2-RL-Math (Cui et al., 2025), while other model families are trained using GSM8K (Cobbe et al., 2021). The downstream performance is evaluated using MATH500 (Hendrycks et al., 2021), AIME 2024 (Li et al., 2024), AMC (Li et al., 2024), OlympiadBench (He et al., 2024), and OMNI-MATH (Gao et al., 2024). For

code tasks, we train the Qwen family and Mistral-24B model using AceCode (Zeng et al., 2025), Eurus-2-RL-Code (Cui et al., 2025), and Kodcode[1].

**Hyperparameters.** For hyperparameters, we use a learning rate of $5 \times 10^{-7}$ for the policy model and $10^{-6}$ for the implicit PRM (Yuan et al., 2025) in PRIME. Both policy and PRMs use a batch size of 256 and a micro-batch size of 128. The rollout stage collects 256 prompts with 8 sampled responses. By default, we set the reference KL divergence coefficient to 0. The $\epsilon$ in policy loss (Equation 4) is 0.2. We filter out prompts that receive all correct or incorrect responses.

### B.2 DETAILED FITTING RESULTS

**Fitting Results on Coding Task.** We present the fitting results of coding task in Figure 7.

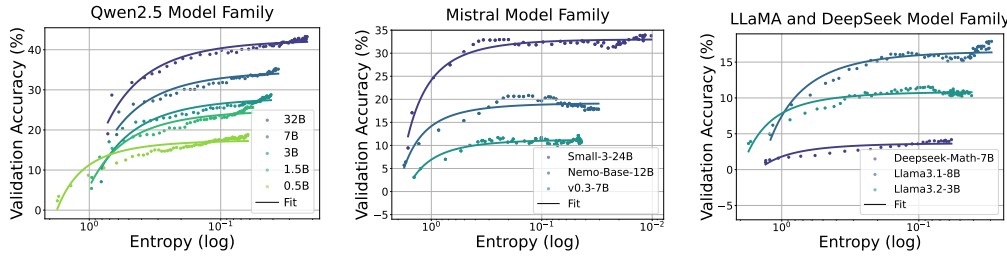

Figure 7: Fitting curves between policy entropy and validation performance in coding task. We conduct validation every 4 rollout steps until convergence.

**Fitting Results of Different Datasets and Algorithms.** In this section, we present more fitting experiment results. The results of training with different datasets and algorithms are presented at Figure 8a and Figure 8b, respectively.

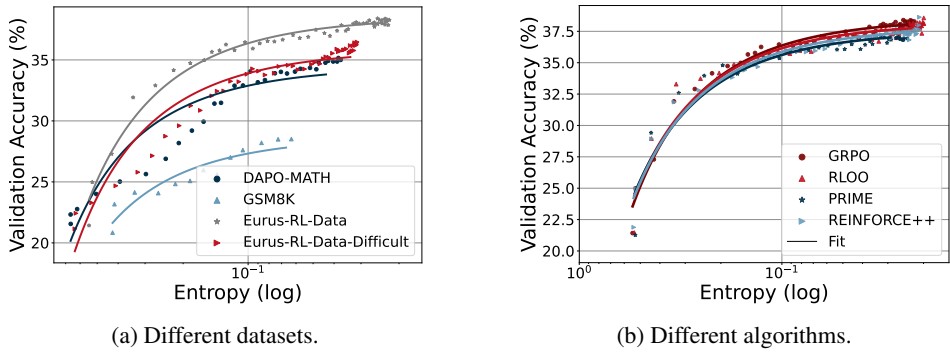

(a) Different datasets.          (b) Different algorithms.

Figure 8: Training Qwen2.5-7B with different datasets and algorithms.

**Fitting Results of Instruct Models.** We also conduct fitting experiments on instruct models, and the fitting function remains valid in our experiments. We present the fitting results in Figure 9.

### B.3 PREDICTING PERFORMANCE FROM POLICY ENTROPY

As we can precisely fit a curve between policy entropy and validation performance, one straightforward application of this fitting is to predict policy performance at low entropy with observations from high entropy data points. To verify that the functional form can be applied at the early stage of RL training, we take a step further by fitting the function within limited training steps and using the fitted function to predict the final performance.

---

[1]We process the data with style instruct and complete into a format that can be handled by unit tests. For the online-judge style, we removed this portion of the data as it was derived from instruct style data.

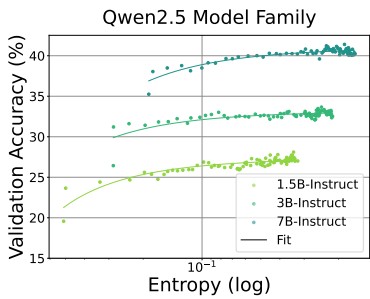

Figure 9: Training Qwen2.5 instruct models on math task.

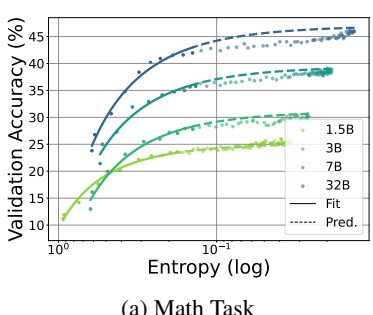

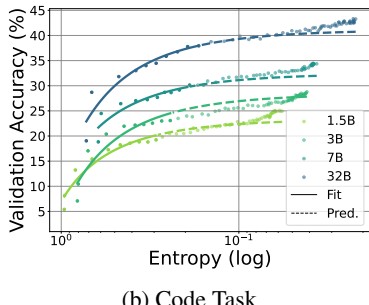

(a) Math Task

(b) Code Task

Figure 10: Predicting the final performance of Qwen2.5 family with only $15\%$ training steps with the fitted function. The average RMSE is $0.9\%$ and $1.2\%$ for all predicted steps, $0.5\%$ and $1.9\%$ for final step performance, respectively.

Take Qwen2.5 family as an example, we fit the function form with coefficients $a$ and $b$ using only the first 36 training steps. Using this function, we perform an advance prediction for the subsequent 200 training steps. As shown in Figure 10, for the math and coding task, we achieve an average Root Mean Square Error (RMSE) of $0.9\%$ and $1.2\%$ during prediction, $0.5\%$ and $1.9\%$ for final performance, respectively. It suggests that the late stage performance of the policy can be estimated early in training, without the need to run the full RL process. Moreover, we can also obtain the final performance of the policy when it becomes static. With $\mathcal{H} = 0$, $R = -a + b$, which is the upper bound of the policy given the training data.

### B.4 UNDERSTANDING THE COEFFICIENTS

**The Coefficients are Algorithm-irrelevant.** We investigate whether different RL algorithms would affect the fitted function. Figure 8b plots the fitted curves with GRPO, RLOO, and PRIME. We find that, although these algorithms apply distinct advantage estimation methods, they do not influence the fitted entropy-performance function. This indicates that the coefficients $a, b$ reflect some intrinsic properties of the policy model and training data.

**Predicting Coefficients when Scaling Parameters.**

Taking a closer look at the coefficients $a, b$, their meanings are clear. By differentiating the equation, we derive $dR/d\mathcal{H} = -a \exp(\mathcal{H})$, which means $a$ is the rate at which the model converts entropy into downstream performance. Also, as stated above, $-a + b$ is the maximum validation score the model can achieve when entropy is fully depleted. Intuitively, $a, b$ should be relevant with model sizes, where larger models could trade entropy for reward more efficiently, as well as achieve higher performance.

To validate this, we again adopt Qwen2.5 model family, since they have similar architecture and undergo similar training process. In Figure 11, we plot the model parameter count (without embedding) versus $a, b$ on math and coding tasks. It is observed that, both $a$ and $b$ vary smoothly with policy size at a log-linear rate. This log-linear relationship between model sizes and coefficients is also

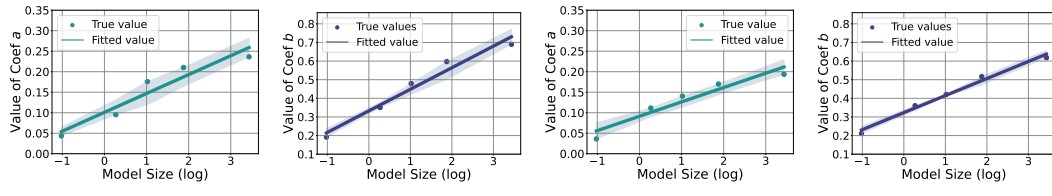

(a) Coef. $a$ for math task    (b) Coef. $b$ for math task    (c) Coef. $a$ for code task    (d) Coef. $b$ for code task

Figure 11: Fitted curves between coefficients and model sizes of Qwen2.5 model family. The model sizes are parameter counts (B) without embeddings. $a, b$ are obtained from experiments in Sec. 2.4. We use log-linear function to fit the curve.

observed in Gao et al. (2022). It allows us to extrapolate the coefficients of larger models based on the training dynamics of smaller models, extending the predictability to the dimension of model sizes. In other words, it enables us to predict the final performance of larger LMs through RL training without actually training them, once we train smaller models within the same family and get their coefficients. Figure 8a also illustrates that the coefficients are related with training data.

## C   PROOF

### C.1   USEFUL LEMMAS

**Lemma 2 (Derivative of softmax function)**

$$\frac{\partial \log \pi_\theta(a \mid s)}{\partial \theta_{s,a'}} = \mathbf{1}\left\{a = a'\right\} - \pi_\theta\left(a' \mid s\right)$$

**Lemma 3 (Expectation of Advantage function given state $s$)**

$$\mathbb{E}_{a\sim\pi_\theta(\cdot|s)}\left[A^{\pi_\theta}(s,a)\right] = \mathbb{E}_{a\sim\pi_\theta(\cdot|s)}\left[Q^{\pi_\theta}(s,a) - V^{\pi_\theta}(s)\right]$$
$$= \mathbb{E}_{a\sim\pi_\theta(\cdot|s)}\left[Q(s,a)\right] - \mathbb{E}_{a\sim\pi_\theta(\cdot|s)}\left[V(s)\right]$$
$$= V(s) - V(s)$$
$$= 0$$

### C.2   PROOF FOR LEMMA 1

**Lemma 1**: *Let the actor policy $\pi_\theta$ be a tabular softmax policy, the difference of information entropy given state $s$ between two consecutive steps satisfies*

$$\mathcal{H}(\pi_\theta^{k+1}|s) - \mathcal{H}(\pi_\theta^k|s) \approx -\mathrm{Cov}_{a\sim\pi_\theta^k(\cdot|s)}\left(\log \pi_\theta^k(a|s), z_{s,a}^{k+1} - z_{s,a}^k\right)$$

*Proof adapted from (Liu, 2025).*

In tabular softmax policy, each state-action pair $(s, a)$ is associated with an individual logit parameter $z_{s,a} = \theta_{s,a}$. We assume that we are updating logits $z$ via $z^{k+1} = z^k + \eta \cdot \nabla J(\pi_\theta)$. Given $\eta$ is relatively small, leveraging Taylor's expansion under first-order approximation, we have

$$\mathcal{H}(\pi_\theta^{k+1} \mid s) \approx \mathcal{H}(\pi_\theta^k \mid s) + \langle \nabla\mathcal{H}(\pi_\theta^k \mid s), (z^{k+1} - z^k)\rangle$$

We then to derive what $\nabla\mathcal{H}(\pi_\theta^k \mid s)$ is, according to the definition of $\mathcal{H}$, we have

$$\nabla_\theta \mathcal{H}(\pi_\theta \mid s) = \nabla_\theta \mathcal{H}(\pi_\theta(\cdot \mid s))$$
$$= \nabla_\theta(-\mathbb{E}_{a\sim\pi_\theta(\cdot|s)}\left[\log \pi_\theta(a \mid s)\right])$$
$$= -\mathbb{E}_{a\sim\pi_\theta(\cdot|s)}\left[\nabla_\theta \log \pi_\theta(a \mid s) + \log \pi_\theta(a \mid s)\nabla_\theta \log \pi_\theta(a \mid s)\right]$$
$$= -\mathbb{E}_{a\sim\pi(\cdot|s)}\left[\log \pi_\theta(a \mid s)\nabla_\theta \log \pi_\theta(a \mid s)\right]$$

Then we have,

$$
\begin{aligned}
\langle \nabla_\theta \mathcal{H}(\theta^k \mid s), (z^{k+1} - z^k) \rangle &= -\langle \mathbb{E}_{a \sim \pi(\cdot|s)} [\log \pi_\theta(a \mid s) \nabla_\theta \log \pi_\theta(a \mid s)], (\theta^{k+1} - \theta^k) \rangle \\
&= -\mathbb{E}_{a \sim \pi(\cdot|s)} [\log \pi_\theta(a \mid s) \langle \nabla_\theta \log_{\pi_\theta}(a \mid s), \theta^{k+1} - \theta^k \rangle] \\
&= -\mathbb{E}_{a \sim \pi(\cdot|s)} \left[ \log \pi_\theta(a \mid s) \sum_{a' \in \mathcal{A}} \frac{\partial \log \pi_\theta(a \mid s)}{\partial \theta_{s,a'}} \cdot (\theta^{k+1}_{s,a'} - \theta^k_{s,a'}) \right] \\
&= -\mathbb{E}_{a \sim \pi(\cdot|s)} \left[ \log \pi_\theta(a \mid s) \sum_{a' \in \mathcal{A}} (\mathbf{1}\{a = a'\} - \pi(a' \mid s)) \cdot (\theta^{k+1}_{s,a'} - \theta^k_{s,a'}) \right] \\
&= -\mathbb{E}_{a \sim \pi(\cdot|s)} \left[ \log \pi_\theta(a \mid s) \left[ (\theta^{k+1}_{s,a} - \theta^k_{s,a}) - \sum_{a' \in \mathcal{A}} \pi(a' \mid s)(\theta^{k+1}_{s,a'} - \theta^k_{s,a'}) \right] \right] \\
&= -\mathbb{E}_{a \sim \pi(\cdot|s)} [\log \pi_\theta(a \mid s)(\theta^{k+1}_{s,a} - \theta^k_{s,a})] + \mathbb{E}_{a \sim \pi(\cdot|s)} \left[ \log \pi_\theta(a \mid s) \cdot \mathbb{E}_{a' \sim \pi(\cdot|s)} \left[ \theta^{k+1}_{s,a'} - \theta^k_{s,a'} \right] \right] \\
&= -\mathbb{E}_{a \sim \pi(\cdot|s)} [\log \pi_\theta(a \mid s)(\theta^{k+1}_{s,a} - \theta^k_{s,a})] + \mathbb{E}_{a \sim \pi(\cdot|s)} [\log \pi_\theta(a \mid s)] \cdot \mathbb{E}_{a' \sim \pi(\cdot|s)} \left[ \theta^{k+1}_{s,a'} - \theta^k_{s,a'} \right] \\
&= -Cov_{a \sim \pi(\cdot|s)} \left( \log \pi(a \mid s), \theta^{k+1} - \theta^k \right) \\
&= -Cov_{a \sim \pi(\cdot|s)} \left( \log \pi(a \mid s), z^{k+1} - z^k \right)
\end{aligned}
$$

## C.3  PROOF FOR PROPOSITION 1

**Proposition 1**: *Let the actor policy $\pi_\theta$ be tabular softmax policy and updated using Eq. 2, the difference of $z_{s,a}$ between two consecutive steps satisfies*

$$
z^{k+1}_{s,a} - z^k_{s,a} = \eta \cdot \pi_\theta(a \mid s) \cdot A(s, a)
$$

*Proof.*

In tabular softmax policy, each state-action pair $(s, a)$ is associated with an individual logit parameter $z_{s,a} = \theta_{s,a}$. Through gradient backtracking, $z_{s,a}$ is updated via $z^{k+1}_{s,a} = z^k_{s,a} + \eta \cdot \nabla_{\theta_{s,a}} J(\theta)$, therefore, we have

$$
\begin{aligned}
z^{k+1}_{s,a} - z^k_{s,a} &= \eta \cdot \nabla_{\theta_{s,a}} J(\theta) \\
&= \eta \cdot \mathbb{E}_{a' \sim \pi_\theta(\cdot|s)} \left[ \nabla_{\theta_{s,a}} \log \pi_\theta(a' \mid s) \cdot A(s, a') \right] \\
&= \eta \cdot \mathbb{E}_{a' \sim \pi_\theta(\cdot|s)} \left[ \underbrace{\frac{\partial \log \pi_\theta(a' \mid s)}{\partial \theta_{s,a}}}_{\text{Lemma } 2} \cdot A(s, a') \right] \\
&= \eta \cdot \sum_{a' \in \mathcal{A}} \left[ \pi_\theta(a' \mid s) \cdot (\mathbf{1}\{a = a'\} - \pi_\theta(a \mid s)) \cdot A(s, a') \right] \\
&= \eta \cdot \pi_\theta(a \mid s) \cdot \left[ (1 - \pi_\theta(a \mid s)) \cdot A(s, a) - \sum_{a' \in \mathcal{A}, a' \neq a} \pi_\theta(a' \mid s) \cdot A(s, a') \right] \\
&= \eta \cdot \pi_\theta(a \mid s) \cdot \left[ A(s, a) - \underbrace{\sum_{a' \in \mathcal{A}} \pi_\theta(a' \mid s) \cdot A(s, a')}_{\text{Lemma } 3} \right] \\
&= \eta \cdot \pi_\theta(a \mid s) \cdot [A(s, a) - 0] \\
&= \eta \cdot \pi_\theta(a \mid s) \cdot A(s, a)
\end{aligned}
$$

## C.4  PROOF FOR THEOREM 2

**Theorem 2**: *Let the actor policy $\pi_\theta$ be tabular softmax policy, and $\pi_\theta$ is updated via natural policy gradient Kakade (2001), the difference of information entropy given state $s$ between two consecutive*

*steps satisfies*

$$\mathcal{H}(\pi_\theta^{k+1}|s) - \mathcal{H}(\pi_\theta^k|s) \approx -\eta \cdot \mathrm{Cov}_{a \sim \pi_\theta^k(\cdot|s)} \left( \log \pi_\theta^k(a|s), A(s,a) \right)$$

*Proof.*

According to Lemma 1, we first derive the difference of logits $z$ in natural policy gradient. We learn from (Agarwal et al., 2021) that, when we are updating policy using natural policy gradient via gradient backtracking, $z_{s,a}^{k+1} - z_{s,a}^k$ satisfies,

$$z_{s,a}^{k+1} - z_{s,a}^k = \eta \cdot A(s,a)$$

Applying this into Lemma 1, we have

$$\mathcal{H}(\pi_\theta^{k+1}|s) - \mathcal{H}(\pi_\theta^k|s) \approx -\eta \cdot \mathrm{Cov}_{a \sim \pi_\theta^k(\cdot|s)} \left( \log \pi_\theta^k(a|s), A(s,a) \right)$$

## D  ADDITIONAL ANALYSIS OF COVARIANCE DYNAMICS

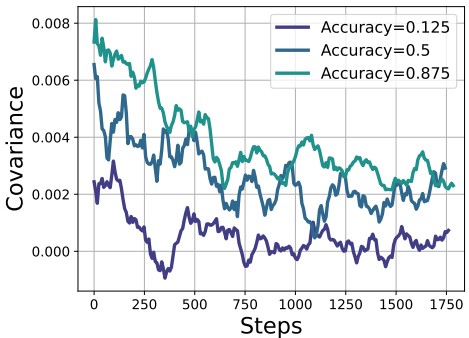

Figure 12: Covariance dynamics across difficulty groups. Easier prompts with higher accuracy show higher covariance, while harder prompts yield smaller covariance.

To further explore the behavior of covariance, we categorize training examples by difficulty using accuracy. As shown in Figure 12, $Cov(\cdot)$ tends to be smaller in magnitude for harder examples, aligning with intuition: when the model struggles to learn, high-probability actions are not reliably associated with higher returns. In contrast, for easier examples, where the model is more confident and calibrated, $Cov(\cdot)$ is larger, indicating stronger alignment between action probabilities and advantage estimates.

## E  EFFECT OF ENTROPY REGULARIZATION

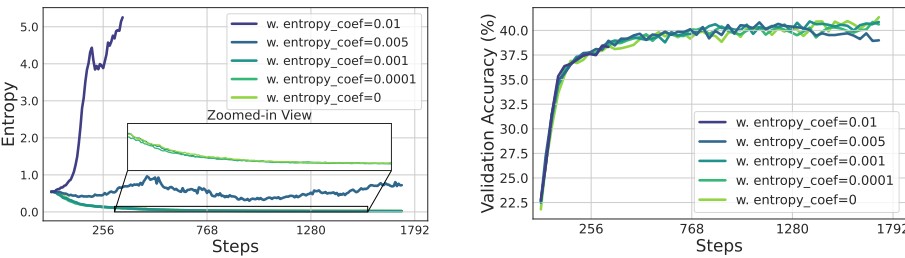

Figure 13: The policy entropy and validation accuracy of adding entropy loss where $L_{\mathrm{ent}} = L - \alpha \mathcal{H}(\pi_\theta)$. $L$ is the original loss and $\alpha$ is the coefficient of entropy loss.

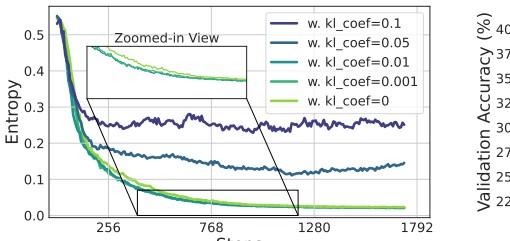 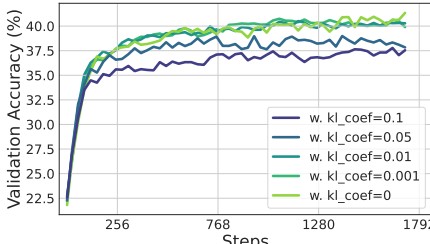

Figure 14: The policy entropy and validation accuracy of adding KL penalty between policy and reference model where $L_{\text{KL}} = L + \beta \mathbb{D}_{\text{KL}}(\pi_\theta || \pi_{\text{ref}})$. $L$ is the original loss and $\beta$ is the coefficient of KL loss.

A common approach in the RL literature to control policy entropy is to apply entropy loss (Schulman et al., 2017b) or KL penalty. We conduct experiments to see whether it is effective for LLMs.

Figure 13 and Figure 14 present the results. It is shown that entropy loss is highly sensitive to coefficients, and it does not outperform other baselines. Despite the reference KL achieves stable entropy values, it fails to improve policy and instead leads to a degradation in performance. Therefore, naively adopting entropy regularization techniques from conventional RL struggles to solve the entropy bottleneck of LLMs. These regularization terms are either hyper-parameter sensitive (He et al., 2025) or degrade policy performance. Therefore, most recent works do not include them as well (Cui et al., 2025; Hu et al., 2025; Liu et al., 2025; Yu et al., 2025).

## F    MORE DETAILS OF THE PROPOSED METHODS

In this section, we present more details of training with our proposed `Clip-Cov` and `KL-Cov`, including the pseudo code and the training hyperparameters.

### F.1    PSEUDO CODE OF THE PROPOSED METHODS

### F.2    TRAINING HYPERPARAMETERS

In each rollout step, we sample 8 responses per prompt for a batch of 256 prompts using a temperature of 1, and subsequently perform 8 policy updates on the collected responses. We also filter out the prompts with all-correct/incorrect responses. The test datasets include MATH500, AIME 2024, AIME 2025 (Li et al., 2024), AMC, OMNI-MATH, OlympiadBench, and Minerva (Lewkowycz et al., 2022). During evaluation, we set the rollout temperature to 0.6 for AIME and AMC, while using greedy decoding for all other test sets. In `Clip-Cov`, the clip ratio $r$ is $2 \times 10^{-4}$, with $\omega_{\text{low}}$ and $\omega_{\text{high}}$ equals 1 and 5, respectively. For `KL-Cov`, the $k$ is set as $2 \times 10^{-3}$ and $2 \times 10^{-4}$ for Qwen2.5-7B and 32B, respectively, the KL coefficient $\beta$ is set as 1. The max generation length is 8192.

```python
def compute_policy_loss(old_log_prob, log_prob, advantages,
    select_ratio, method, **args):
    ratio = exp(log_prob - old_log_prob)
    pg_losses1 = -ratio * advantages
+   # calculate token wise centered cross - product
+   covs = (log_prob - log_prob.mean()) * (advantages - advantages.
    mean())
+   select_num = int(select_ratio * len(pg_losses1))
    if method == "clip_cov":
        pg_losses2 = -clip(ratio, args["clip_range_lb"], args["
            clip_range_ub"]) * advantages
+       # randomly select index to be detached
+       clip_idx = random_select(covs[covs > args["cov_lb"] & covs <
        args["cov_ub"]], num=select_num)
+       pg_losses1[clip_idx].detach_()
+       pg_losses2[clip_idx].detach_()
        pg_loss = maximum(pg_losses1, pg_losses2).mean()
    if method == "kl_cov":
        kl_coef = args["kl_coef"]
        kl_penalty = (log_prob - old_log_prob).abs()
-       pg_losses = pg_losses1 + kl_coef * kl_penalty
+       # find out index with highest conviriance
+       select_idx = topk(covs, k=select_num, largest=True)
+       # apply KL penalty of these samples
+       pg_losses1[select_idx] += kl_coef * kl_penalty[select_idx]
        pg_loss = pg_losses1.mean()
    return pg_loss
```

Algorithm 1: The pseudo-code of the policy loss computation with `Clip-Cov` and `KL-Cov`. The implementation only need to modify several lines of code.

