# OpenReview forum: "The Entropy Mechanism of Reinforcement Learning for Reasoning Language Models"
_ICLR.cc/2026/Conference — ICLR 2026 Conference Withdrawn Submission_

### Official Review · Reviewer_7E3R · 2025-10-26

**Soundness:** 2
**Presentation:** 3
**Contribution:** 1
**Rating:** 2
**Confidence:** 3

**Summary:**

The authors find an interesting relationship between policy entropy and performance in LLMs trained with RL. This shows that they are traded offs, that is, an increase in policy entropy amounts to diminish increases in performance, suggesting that maintaining an optimal level of policy entropy would be important. Indeed, the authors finds through experiments that restricting updates to high covariance tokens improves performance because policy entropy is sustained in this case.

**Strengths:**

The main strength of the paper is making a good diagnosis for why performance in LLMs does not increase over time and rapidly saturates. Identifying that this is most likely due to the collapse of policy entropy is important.

**Weaknesses:**

The main weakness is that the empirical evidence is not proof that entropy collapse is the actual reason for lack of performance improvement, so it remains to be seen whether this is the main reason.

Although the authors connect policy entropy changes with the covariance between the log policy and the logits differences, math is weak and subject to strong approximations. For instance, in Lemma1, the actual policy entropy difference should include in the first term an average over the new distribution over states $s$ based on $d_{\theta_{k+1}}$, not over $d_{\theta_{k}}$. Ignoring this dependence neglects a contribution to the difference with is of the same order as the covariance term, obviating half of the effects.

Most of the propositions are trivial, like Proposition 1, or based on approximation but without providing bounds or stablishing > o < relationships.

Performance increase seems small or not present in some cases compared to baselines (Table 2).

**Questions:**

Can the authors provide a full approximation to the actual difference for the policy entropies, not neglecting the new state distribution?

The extrapolation of Fig. 10 does not seem very impressive: the prediction lies on the flat side of the curve, so it is not surprising that the fit is good. Can the author reverse the predictions, that is, fit the last part of the curve and see how the early curve is predicted by the fit?

Line 971: missing $\theta$ in the policy average of last equation

Line 984: two full lines are repeated.

Why is the last term of eq. in line 985 zero?

---

### Official Review · Reviewer_HRYa · 2025-10-31

**Soundness:** 3
**Presentation:** 3
**Contribution:** 2
**Rating:** 4
**Confidence:** 3

**Summary:**

The paper studies the phenomenon in RL-for-reasoning LLMs: policy entropy crashes to $0$ quickly, and at exactly the same time the validation accuracy stops improving. The paper establishes an empirical observation that $R= -a \exp \mathcal{H} + b$ between entropy $\mathcal{H}$ and downtream performance $R$. To increase exploration, motivated by this observation, the authors propose two techniques that apply KL penalty to tokens with high covariances. The authors conduct experiments to show effectiveness of the performance.

**Strengths:**

* The paper is well-organized and clearly written. The paper has clean and good illustrations for their takeaways and empirical findings.
* The empirical findings support the two methods proposed by the authors, that suppressing tokens with high logit-advantage covariances performs better.

**Weaknesses:**

Major comments:
* I apologize for my unfamiliarity with the empirical studies---however, it strikes to me that it should be well known: for both finite‐ and infinite‐horizon Markov decision processes--discrete or continuous state/action spaces--if the system is fully observable and the Bellman operator satisfies standard measurability and compactness conditions, then there exists an optimal deterministic stationary policy. For example, see Puterman, Martin L. (1994). Therefore, the empirical observation of "collapsing entropy" shouldn't be surprising, and in fact well-known in the theoretical world. While this does constrain exploration, methods such as NPG are proposed exactly to deal with this issue by smoothing optimization landscape, see Cen et al. (2022). This seems very much aligns with the Clip-Cov and KL-Cov algorithms. Though, I admit the end-goal here is different, and this paper is more empirically inspired.
* The empirical reward function is also somewhat not surprising---since the entropy are logit probabilities, taking exponential really recovers the probability gap between the policy and the optimal deterministic policy---which is exactly the error rate. Overall, I think the entropy mechanism is a bit overblown from my background in RL theory. I'm open to further discussions.

Puterman, Martin L. "Markov Decision Processes: Discrete Stochastic Dynamic Programming." (1994).
Cen, Shicong, et al. "Fast global convergence of natural policy gradient methods with entropy regularization." Operations Research 70.4 (2022): 2563-2578.

Minor comments:
* P7, Line 328. "..struggles.." should be "..struggle.."

**Questions:**

* My main questions are in the weaknesses section.
* Can the authors explain a bit more what is the performance gain from? I think it is mainly from the model misspecification (as it's not really a tabular RL) and overparameterization. The locally optimal policy is perhaps not unique. Having a larger exploration capacity allows the algorithm to escape into a better landscape of a better local minima. If this is really the case, perhaps the authors should discuss a bit more on this insight.

---

### Official Review · Reviewer_Bebi · 2025-11-01

**Soundness:** 3
**Presentation:** 3
**Contribution:** 3
**Rating:** 6
**Confidence:** 4

**Summary:**

This paper aims to address a key obstacle in scaling reinforcement learning for reasoning tasks with large language models: policy entropy collapse. The paper's contributions are threefold: (1) An empirical finding that policy entropy drops sharply, leading to performance saturation, and is governed by a concise exponential relationship $R = -a \cdot \exp(\mathcal{H}) + b$. (2) A theoretical analysis explaining this collapse, deriving that entropy dynamics are driven by the covariance between an action's log-probability and its advantage ($Cov(log(\pi), A)$). (3) Two simple and effective regularization techniques, Clip-Cov and KL-Cov, which are based on this theoretical insight and manage entropy by restricting updates on high-covariance tokens. Experimental results show these methods maintain entropy, enable continuous exploration, and achieve superior performance on math reasoning benchmarks.

**Strengths:**

i. The paper presents a potentially impactful analysis of the **"entropy collapse"** problem in LLM-RL. The "performance-entropy" exponential relationship and the "covariance-driven" theoretical mechanism are important scientific contributions to the field. Furthermore, the proposed Clip-Cov/KL-Cov methods are principled solutions based on this insight, and their proven effectiveness provides a new approach to solving the exploration problem in RL.

ii. The paper delves deep into why existing methods fail, providing a complete answer that covers **"what" (the empirical law), "why" (the theoretical mechanism), and "how" (the new algorithm)**.

**Weaknesses:**

### Weaknesses:
While the paper is strong, there are several weaknesses that need to be addressed:

(1) Limitation of Theoretical Assumptions: The core theory (Lemma 1, Prop 1, Thm 1 & 2) is derived based on a "tabular softmax policy." This is a very strong simplifying assumption. The gap from a tabular setting to a large-scale Transformer is enormous, and the authors do not sufficiently discuss why this tabular-based derivation applies so well to complex function approximation (FA).

(2) Inconsistency in Covariance Granularity: There appears to be an inconsistency in the definition of "covariance" throughout the paper: the theory (Thm 2) uses an action-level expectation, the empirical validation (Eq. 7) uses a sequence-level computation, and the algorithm (Eq. 8) operates at the token level. The link between these three granularities is not clear, especially why the token-level proxy is effective for controlling the sequence-level entropy.

(3) Completeness of Experimental Baselines: Appendix E briefly dismisses standard entropy regularization and KL penalty, but the results (Figs 13 & 14) appear undertuned. A key question is whether the GRPO baseline included the standard PPO entropy bonus. If not, the improvement from Clip-Cov/KL-Cov might simply be because any entropy regularization is better than none. The paper needs a clearer comparison against a well-tuned standard entropy-regularized PPO.

(4) Hyperparameter Sensitivity: The paper notes that the ratio $k$ for Clip-Cov and KL-Cov is very small ($10^{-4}$ to $10^{-3}$). This seems to be a very sensitive and potentially hard-to-tune hyperparameter. Figure 6 demonstrates controllability but also implies that different $k$ values lead to drastically different entropy dynamics.

### Minor comments:
In Figure 1 (left), the call-out lines and annotations (e.g., "95% improvement") are slightly chaotic and hard to map to the curves. I recommend clarifying these annotations in the final version for better readability.

**Questions:**

To be more confident in my final assessment, I would like the authors to clarify the following points:
1. Could you elaborate on why you believe the theoretical derivation based on a "tabular softmax" policy can be successfully transferred to explain phenomena observed in large-scale Transformer models? Under function approximation, why does this covariance mechanism remain qualitatively the same?
2. As stated in Weak Point 2, how do you bridge the gap between the "action" covariance (Theory), the "sequence" covariance (Empirical Validation), and the "token" covariance (Algorithm)? Specifically, why is computing $Cov(log(\pi_t), A_t)$ at the token level still a correct or effective proxy for controlling the sequence-level entropy?
3. In your baseline GRPO experiments (e.g., Figure 5), did you include the standard entropy bonus term (i.e., from PPO)? If so, what was its coefficient? If not, could you justify its omission and provide a comparison against a baseline with a standard (and well-tuned) entropy bonus?
4. Your method appears highly sensitive to the clipping/penalizing ratio $k$. Is this difficult to tune in practice? Could you provide a robustness analysis for the choice of $k$? Furthermore, does the random_select in Algorithm 1 introduce significant training variance?

---

### Official Review · Reviewer_CByb · 2025-11-02

**Soundness:** 1
**Presentation:** 2
**Contribution:** 2
**Rating:** 2
**Confidence:** 4

**Summary:**

+ Summary & Contributions
	- The authors focus their attention on the loss or collapse of policy entropy during the course of RL fine-tuning.
	- To foster insight into the entropy collapse issue, the authors propose fitting an equation to model the relationship between policy entropy and downstream LLM performance.
	- Further theoretical analysis by the authors establishes a relationship between the change in policy entropy between successive updates and the covariance of policy log-probabilities with the change in policy logits. For specific policy-gradient methods, this can be shown to simplify (under some technical assumptions) to the covariance between the log probability of a given state-action pair under the current policy and the advantage of the state-action pair.
	- These theoretical results motivate an empirical approach for combatting entropy collapse through the regularization of high-covariance actions, either by clipping or KL-regularization. Experimental results confirm that the proposed approach is effective at preserving response distribution entropy over the course of RL fine-tuning while also yielding higher test accuracy relative to standard GRPO.

**Strengths:**

+ Quality
	- Strengths
		- The authors seem to offer a novel theoretical perspective on the entropy collapse phenomenon. The technical proofs are mostly correct, except for an issue in results the authors have borrowed discussed below.
		- The resulting covariance-based regularization scheme studied empirically by the authors seems like a simple, reasonable approach to try in light of the presented theoretical findings.
	- Weaknesses
		* Major
			- It seems quite underwhelming that the exponential relationship established by the authors between policy entropy and performance only holds in the absence of entropy/KL-regularization; these are two standard heuristics that are near-ubiquitously used by policy-gradient methods, particularly in RL fine-tuning because of these well-known entropy collapse issues. In the absence of those regularization methods, the findings of Figure 1 (left) are perhaps to be expected and could likely even be obtained in standard deep RL environments. What is the value of the $R = -a \exp(\mathcal{H}) + b$ equation when it only holds under conditions not resemblant of how RL fine-tuning of LLMs is current performed?
			- While the brief, informal presentation of the exponential relationship between policy entropy and performance was fine for the abstract and introduction, I had expected a proper, formal presentation of what the authors had in mind by the time Section 2.3 came around. Unfortunately, the authors never actually define $R$ and also leave the ambiguous $\mathcal{H}$ for policy entropy. Is the latter defined according to Equation 5? Why not use that notation or some variant of it to state things properly? How is $R$ being defined? More importantly, this exponential equation seems largely inconsequential to the theory and subsequent entropy-preserving approach studied empirically by the authors. The entropy collapse that occurs when fine-tuning LLMs is well-documented and belaboring that point with this exponential function actually seems like a waste of space in the main paper that could be better allocated towards ideas from Section 3 onwards.
		* Minor
			- It would be nice for the authors to not blindly reference a long JMLR paper with numerous theoretical results for policy-gradient methods, Agarwal et al., (2021), and actually indicate which result(s) they are utilizing in this work. In the first step of the proof for Theorem 2, it seems the authors have either missed or dropped a $\frac{1}{(1-\gamma)}$ term when borrowing Lemma 15 of Agarwal et al., (2021). This raises a broader point that the authors are utilizing a finite-horizon MDP setting in this work whereas the results of Agarwal et al., (2021) are for the infinite-horizon, discounted MDP setting. There is likely a bit of translation work needed to port over those results and some problem-specific constants are missing.

+ Clarity
	- Strengths
		- Overall, the paper is well-written and reasonably structured.
	- Weaknesses
		* Major
			- N/A
		* Minor
			- The authors seem to contradict themselves in the abstract when discussing "a rare action with high advantage would increase policy entropy" (L24-25) and after Theorem 1 "action a receives both low probability and low advantage would lower the entropy". I believe the latter is correct and what the authors want to emphasize since the former would be the logical conclusion just by examining the advantage actor-critic update rule.


+ Originality
	- Strengths
		- To the best of my knowledge, a theoretical result connecting the policy entropy between successive policy-gradient updates and covariance is novel.
		- To the best of my knowledge, an empirical approach that selectively applies either clipping or KL-regularization based on the covariance of log-probabilities and advantages is novel.
	- Weaknesses
		* Major
			- The authors are not the first to observe the entropy collapse phenomenon and feel compelled to take steps empirically to address it. The Related Work section seems rather incomplete in identifying alternative approaches to dealing with entropy collapse. For example, [1] also give a theoretical characterization of how entropy decreases with successive policy-gradient updates and offers a resolution based on interpolation of fine-tuned and base model policy parameters. More broadly, the authors could do with a much more comprehensive overview of alternative approaches to entropy collapse resolution and the trade-offs to consider between those methods and their proposed approach.
		* Minor
			- N/A

+ Significance
	- Strengths
		- A simple, easy-to-implement heuristic for consistently preserving policy entropy and retaining diversity of the response distribution would likely be a welcome and impactful contribution.
	- Weaknesses
		* Major
			- Throughout the entire paper, there doesn't seem to be a single mention of how many random trials/seeds are used to obtain the reported results. Additionally, not one plot or table appears to include some notion of standard error/confidence that communicates whether or not the proposed approach consistenly delivers performance improvements. Consequently, it is premature to claim Table 1 reports "best results" in boldface text without actually knowing if they are indeed statistically significant.
			- If the number of trials used in the experiments is sufficiently large and the results are statistically significant, the authors have only succeeded in demonstrating the bare minimum that their proposed approach surpases GRPO; notably, GRPO is but one burgeoning method for RL fine-tuning of LLMs, the other being PPO. Why are there no PPO comparisons? More to the point, as the authors argue, neither standard PPO nor GRPO make a concerted effort to preserve policy entropy. Where are the comparisons to other baseline methods that take alternative perspectives and methodologies to mitigating entropy collapse?
		* Minor
			- N/A


+ Final Remarks
	- While I have identified issues on the axes of quality and originality, the main issues working against this paper are those of significance. Namely, it is entirely unclear if the reported empirical results are statistically significant or if the proposed approach represents a meaningful advance over existing approaches for preserving policy entropy and retaining response distribution diversity. I have provided a citation to one method I know of, but I would strongly suspect there are likely many more approaches the authors have failed to acknowledge in their literature review; while comparing against all of them is unrealistic, comparing against none of them is also unacceptable. With that in mind, it doesn't seem that the paper is ready for publication at this time.


+ References
	1. Veselovsky, Veniamin, Benedikt Stroebl, Gianluca Bencomo, Dilip Arumugam, Lisa Schut, Arvind Narayanan, and Thomas L. Griffiths. "Hindsight Merging: Diverse Data Generation with Language Models." In The 41st Conference on Uncertainty in Artificial Intelligence.

**Weaknesses:**

Please see above.

**Questions:**

Please see above.

---

### Note · Authors · 2025-12-04

I have read and agree with the venue's withdrawal policy on behalf of myself and my co-authors.